ecology/biophysics

restriction–modification systems, bacteriophage, biodiversity, predator partitioning, growth–defence trade-off

**Author for correspondence:**
Sandeep Krishna
e-mail: sandeep@ncbs.res.in

# Defence versus growth in a hostile world: lessons from phage and bacteria

## Rasmus Skytte Eriksen[1] and Sandeep Krishna[2]

[1]Niels Bohr Institute, University of Copenhagen, Copenhagen, Denmark
[2]Simons Centre for the Study of Living Machines, National Centre for Biological Sciences, Tata Institute of Fundamental Research, Bangalore, India

RSE, 0000-0002-5860-3838; SK, 0000-0002-0581-173X

Bacterial communities are often highly diverse with several closely related species (or strains) coexisting together. These bacteria compete for resources and the competitive exclusion principle predicts that all but the fastest-growing bacteria will go extinct. When exposed to phage, it is predicted that bacterial strains with restriction–modification (RM) systems can circumvent the competitive exclusion principle and reach diversity of the order of the phage burst size. We show that with a trade-off between bacterial growth rates and the strength of their RM systems, the diversity of such an ecosystem can further increase several fold beyond the burst size limit. Moreover, we find that the ratio of the growth rate of a bacterial strain to the imperfection of its RM system is an excellent predictor of (i) whether the strain will go extinct or not, and (ii) the biomass of the strain if it survives. In contrast, the growth rate alone is not a determinant of either of these properties. Our work provides a quantitative example of a model ecosystem where the fitness of a species is determined not by growth rate, but by a trade-off between growth and defence against predators.

# 1. Introduction

Bacteria and their phage predators are the most abundant organisms on Earth and are present in almost every known ecosystem. Due to their ubiquitous presence, bacteria strongly influence local environments, while phages in turn strongly influence the composition of the bacterial population as well as the selection pressures on individual bacterial strains [1,2]. It is therefore important to understand the dynamics of phage–bacteria ecosystems. A key feature of such ecosystems is the constant co-evolution of bacterial defence systems against phage, and phage mechanisms to avoid or bypass these defences [3–6]. Here, we focus on the defence provided by

restriction–modification (RM) systems in bacteria. These consist of a pair of enzymes, which target the same specific DNA sequences (typically 4–6 bases long). The restriction enzyme cuts the DNA at these sites if they are unmethylated, whereas the corresponding methyltransferase modifies the DNA at these sequences, thus protecting it from the restriction activity. If a phage enters a bacterium with such an RM system, and it is not methylated at the target sequences, it will be destroyed with a high probability. However, there is some chance that it will be methylated at all the sites by the methyltransferase before the restriction enzyme can act, thus escaping the RM system to proliferate unhindered. We will refer to the probability with which an RM system successfully destroys an unprotected infecting phage as its 'efficacy' or 'strength'.

RM systems are found in 90% of all bacterial genomes [7], and also occupy a significant portion of bacterial genomes, on average even more than CRISPR defences [5]. This is curious because at first glance RM systems appear to be a very weak defence against phage. This is because if even a single phage particle manages to evade the defence by becoming appropriately methylated before it is destroyed by the restriction enzyme, then it and all of its future offspring will be immune to this RM system. The probability that any given phage evades the defences is in the range approximately $10^{-2}$–$10^{-8}$ [8,9]. Given that phage burst sizes are of the order of hundreds [10], even with highly effective RM systems a population of bacteria will eventually let a phage through its defences. However, in an ecosystem consisting of many different strains of bacteria with different RM systems, it turns out that this apparently transient defence can result in effective cooperation between bacterial strains and a long-term boost in diversity [11–13]. A previous mathematical model [13] has argued that the burst size of the phage sets an upper bound on the diversity of bacterial strains—the number of strains that can coexist. In contrast, in the absence of phages, competitive exclusion [14,15] would allow only a single strain to survive in the long run. The burst size sets a limit to the diversity in the presence of phage due to a combination of two reasons [13]: (i) in the presence of a phage predator the bacterial strains effectively cooperate rather than compete with each other, resulting in the biomass of each strain being almost identical, irrespective of the intrinsic growth rate; and (ii) for an ecosystem of coexisting bacteria to be stable, on average only one phage from every burst (of size $\beta$) should successfully infect a bacterium and produce a new burst. If the diversity, i.e. number of coexisting strains, is $D$, then the biomass of each strain is proportional to $1/D$ and the number of phage from a particular burst that will on average infect a bacterium with an RM system they can bypass is $\beta/D$. Thus, if the diversity exceeds the burst size, the number of successful phage from each burst will be less than 1 and by argument (ii) the system will not be stable—the phage will die out, followed by extinction of most bacterial strains due to competitive exclusion.

This previous model does not, however, consider the correlation between the efficacy of an RM system and its cost, in terms of reduction of the growth rate of the bacterium. Rather, this model assumes the cost and efficacy are uncorrelated, which allows the appearance in the model of unrealistic, near cost-less near-perfect RM systems. Presently, the relationship between the cost and efficacy of RM systems has not been determined with quantitative precision. However, it is known that such costs exist, although they are small—RM systems with high efficacy seemingly impose only a few per cent penalty on the growth rate for bacteria. Such costs can be observed in competition assays between strains with and without RM systems [9,16], but are hard to detect in bacterial doubling rates [9]. Mechanistically, RM defences have been speculated to give a fitness cost [7], e.g. some RM systems require large amounts of ATP during restriction. It has also been shown that some RM systems may act as autoimmune systems that cause DNA damage on the strains containing them [9]. These costs seemingly imply that RM systems impose a cost on their host which is, at least partially, correlated to their efficacy.

In this paper, we study the consequences of such cost–efficacy correlations in a mathematical model. Because quantitative measurements of these correlations are as yet very few, our work is exploratory in nature. We examine a range of scenarios, testing different functional forms for the underlying correlations and trade-offs between cost and efficacy ranging from a weak trade-off (with typical RM systems decreasing the growth rate by a few per cent, as inferred from competition experiments) to a much stronger trade-off where typical RM systems cut the growth rate in half. We find that surprisingly high diversities are possible—the number of bacterial strains that can coexist can be several-fold larger than the burst size of the phage. We argue that such high diversity is possible because the phage–bacteria interaction, in this case, allows even bacteria with quite low growth rates, and therefore low steady-state biomass, to stably coexist with bacteria with much higher growth rates and biomass.

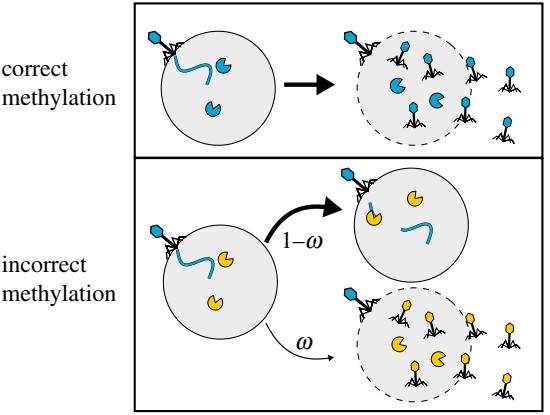

**Figure 1.** Illustration of phage infections. The fate of the phage infection depends on the methylation state of the phage genome. If all recognition sites on the phage genome have been methylated, the RM system is ineffective and the phage will ultimately lyse the bacterium. If the phage does not have the methylated recognition sites, the RM system will almost always cleave the phage genome and prevent the proliferation of the phage, but with probability $\omega_i$ the phage be able to escape the RM systems and lyse the bacterium. Note that the progeny phage will now be methylated differently than the parent phage.

# 2. Methods and materials

## 2.1. An ecosystem of phage and bacteria with restriction–modification systems

We consider an ecosystem consisting of one type of phage and $N$ bacterial strains all of which are potential hosts for the phage but who contain different, unique, RM systems. Due to the presence of different RM systems, there exists an equal number, $N$, of epigenetic variants of the phage which are appropriately methylated to avoid the corresponding RM systems. In a well-mixed ecosystem of this sort, the equations describing the dynamics of the densities of each bacterial strain, $b_i$, and each epigenetic phage variant, $p_i$, are (see [13]):

$$\dot{b}_i = \gamma_i b_i (1 - B/C) - \eta_i b_i p_i - \eta_i \omega_i b_i (P - p_i) - \alpha b_i \tag{2.1}$$

and

$$\dot{p}_i = \eta_i \beta_i b_i p_i + \eta_i \omega_i \beta_i b_i (P - p_i) - \eta_i p_i B - \delta_i p_i. \tag{2.2}$$

These equations model growth of bacteria, phage adsorption followed by either cell lysis and a burst of offspring phage or the destruction of the phage by the RM system (figure 1), as well as the death of phage and bacteria. Here, $\gamma_i$ is the growth rate of the $i$th bacterial strain, $B = \sum_i b_i$ is the total bacterial density, $C$ is the total bacterial carrying capacity of the environment, $P = \sum_i p_i$ is the total phage density, $\alpha$ is the rate at which bacteria are diluted from the ecosystem, and $\omega_i$ is the imperfection of the $i$th RM system, i.e. the probability that a phage that is *not* correctly methylated will escape the RM defence. $\omega$ is thus inversely related to the efficacy of the RM system. The rest of the parameters describe phage properties: $\eta_i$ is the rate of adsorption to the $i$th bacterial strain, $\beta_i$ is the burst size, i.e. the number of phage progeny released during lysis of the $i$th bacterial strain, and $\delta_i$ is the rate at which the $i$th phage variant decays. Various choices can be made for these parameters—we will return to this point in a moment. The only difference from the equations of [13] is that we explicitly include the carrying capacity $C$, which they took to be unity. This does not in fact change the dynamics because if we simply measure all numbers in units of the carrying capacity, then we recover the equations from [13]. That is, our equations exhibit exactly the same dynamics as equations where $C$ is replaced by 1, $\eta$ by $\eta C$, and all populations $b_i$, $p_i$ by $b_i/C$ and $p_i/C$.

## 2.2. Open system with periodic addition of new bacterial strains

With a fixed number of bacterial strains, the bacterial and phage densities typically reach a steady state after order 10 bacterial generations [13]. However, over even longer timescales, one may expect that new strains of bacteria (with different, unique RM systems) arise by invasion, mutation, or acquisition of RM systems from the environment or other bacteria. To model this, we begin with an ecosystem consisting of $M$ strains of bacteria and the corresponding $M$ phage variants, and periodically, at times $T, 2T, 3T, \ldots$,

introduce a new bacterial strain with a unique RM system along with the corresponding phage variant, both at a very low densities corresponding to a single individual. $T$ is typically chosen large enough to allow the system to reach steady state between additions of new strains. At the same time as new species are added, we remove all bacterial strains and phage variants whose density has fallen below the threshold density of 1. Again, we emphasize that the dynamics would be the same whether we choose $C = 10^8$ and this threshold to be 1, or $C = 1$ and this threshold to be $10^{-8}$ (as was used in [13]). Our choice makes it a little more transparent that we are working in the large population limit. If $C$ were made much smaller, without changing the extinction threshold, then we would be in the small population regime where it would be more appropriate to use stochastic versions of our equations.

## 2.3. Choice of parameter values and correlation between cost and efficacy of restriction–modification systems

In [13], the phage parameters $\eta$, $\beta$ and $\delta$ were chosen to be the same for all phage variants, whereas $\omega_i$ and $\gamma_i$ were allowed to vary between strains. We do the same (see electronic supplementary material, table S1 for a full list of parameter values), except that we choose the $\omega_i$ and $\gamma_i$ values differently. Reference [13] sampled $\omega_i$ from a log-uniform probability distribution between $10^{-4}$ and 1, and the $\gamma_i$ independently from a uniform probability distribution between $\alpha$ and 1. We will refer to such sampling as the 'uncorrelated RM system' case. We deviate slightly from [13] and sample the growth rates $\gamma_i$ uniformly between 0 and 1. We also examine the 'correlated RM system' case, where we draw the pair $(\gamma_i, \omega_i)$ from a joint probability distribution $\Pr(\gamma, \omega)$, chosen so that the two quantities are correlated, with an increased strength of the RM system resulting in an increased cost to the bacterium via a decreased growth rate. We compute the $\gamma$ ($\omega$) values by taking a product of appropriately scaled (log) uniform random numbers. The correlations arise because the scalings applied to the random numbers are controlled by a common parameter (see electronic supplementary material, section S2 for details). Wherever we compare the correlated with the uncorrelated RM case, we make sure that the correlated distribution is normalized such that the (marginal) expectation values for the growth rates and RM strengths are the same as in the uncorrelated distribution (i.e. $\langle\gamma\rangle = 0.5$, and $\langle\log10(\omega)\rangle = -2$).

Figure 2a schematically exhibits the cost–efficacy correlations in our model. Here, the unique RM systems are shown in different colours, and the corresponding epigenetic phage variants are correspondingly coloured. The barred arrows connecting the phage to the primary hosts indicate that phage variants strongly suppress the bacterial strain with the corresponding RM systems but only weakly suppress the other strains. The correlations result in strains with weak RM systems (less pac-men) reproducing faster, and vice versa.

For the correlated RM system, we investigate how important the structure of the joint distribution $\Pr(\gamma, \omega)$ is in two ways: (i) We change the 'steepness' of the distribution, i.e. how much the efficacy of an RM system reduces the growth rate of a bacterium, and (ii) we change the Pearson correlation coefficient of the distribution while keeping the marginal means constant. For these two purposes we use two different choices of $\Pr(\gamma, \omega)$ which are parametrized differently to make it easier to change the steepness or the correlation coefficient.

In the first case, we decrease the costs of the RM systems (see electronic supplementary material, section S2 for the details). Essentially, we scale the $\gamma$ values which allows us to control the steepness of the joint probability distribution in the $\gamma$–$\omega$ plane (quantified as the slope $a$ determined by a least-square fit to $\gamma = a\log10(\omega) + b$).

The above tuning of the steepness does result in a small change to the correlation coefficient. Therefore, to understand how changing the correlation coefficient affects out results, we generate $(\gamma, \omega)$ pairs by sampling from a truncated multivariate normal distribution. This new distribution has the property that the Pearson correlation coefficient, $\rho$, between $\gamma$ and $\omega$ is directly controllable. To compare with the previous distributions, we normalize the multivariate normal distribution such that the pairs generated have the same marginal expectation values as for the uncorrelated and correlated distributions above. See electronic supplementary material, section S2 for reference.

## 2.4. Solution to the dynamical equations

In electronic supplementary material, section S3, we solve equation (2.2) for the steady state and we construct a simplified method for determining the diversity of the system. The steady-state analysis produces: (i) equations for the total bacterial density, $B$, and the total phage density, $P$, in terms of the

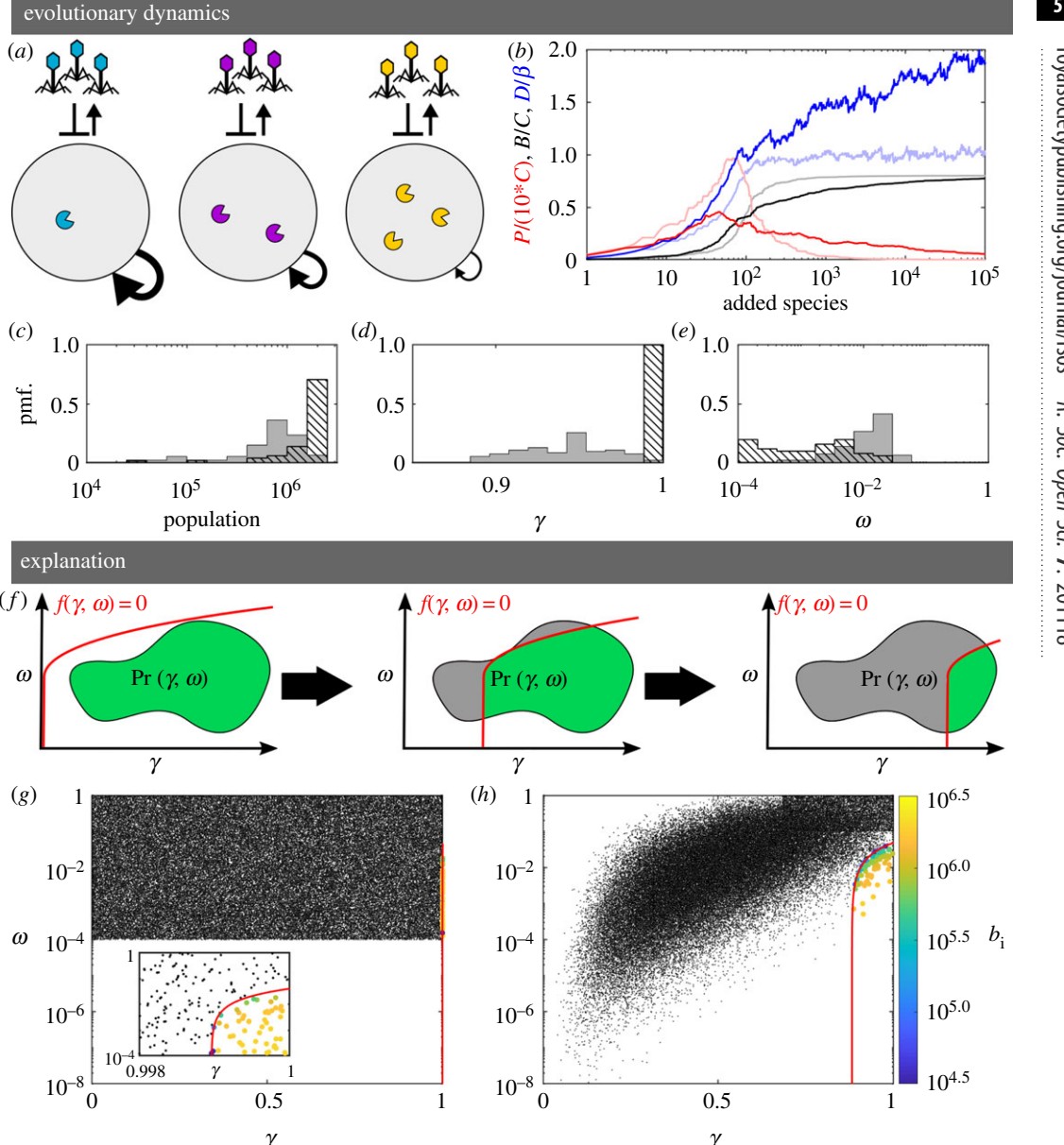

**Figure 2.** Correlations and diversity. (*a*) Schematic of the model. The system consists of populations of bacterial strains with different, unique RM systems, each represented by a colour. Each of the bacterial strains reproduces at a rate $\gamma_i$, which is inversely proportional to the strength of the RM system it has (correlated case). For each such strain, there is a corresponding phage variant (with matching colour) which has the pattern of methylation required to avoid that RM system. Phage with wrong patterns of methylation can only infect a bacterium with probability $\omega_i$. (*b*) Time courses from a simulation with correlated RM systems showing the total bacterial density, $B$, and the total phage density $P$, in units of the carrying capacity $C$. Also shown is the diversity $D$ (the number of coexisting bacterial strains) in units of the phage burst size $\beta$. The dark (light) colours correspond to the case of (un)correlated RM systems. (*c*–*e*) Distributions at the end of the simulation. (un)correlated RM systems shown as solid (hatched) bars. (*c*) Distribution of strain population sizes. (*d*) Distribution of growth rates. (*e*) Distribution of RM imperfections. (*f*) Schematic of the simplified model in $\gamma$–$\omega$ space. The red line indicates the boundary between viable and non-viable strains (representing equation (2.3) in the main text). The filled area (grey+green) indicates the region from which ($\gamma$, $\omega$) pairs are drawn. The green region depicts the portion of this area where species are viable. As species are added, the region corresponding to viable strains constricts. (*g*) Black dots indicate the available strains from panel (*b*, light curves, uncorrelated). Coloured points indicate the species surviving at the end of the simulation. Colour scale is proportional to log10 of the population. Inset zooms in on the relevant region. (*h*) Same as (*g*) but for correlated RM systems.

set of $\omega_i$ and $\gamma_i$ values (see electronic supplementary material, equations (S21) and (S14)); (ii) an equation for each individual bacterial density, $b_i$, also in terms of the set of $\omega_i$ and $\gamma_i$ values. From the latter, (S22), we observe that in order for *all* the $b_i$s to be positive in steady state, each pair ($\gamma_i$, $\omega_i$) must fulfil:

$$f(\gamma_i, \omega_i) = \gamma_i(1 - B/C) - \alpha - \eta\omega_i P > 0. \tag{2.3}$$

Thus, given the set of $\omega_i$ and $\gamma_i$ values, we can calculate which bacterial strains will go extinct once the system reaches steady state.

### 2.4.1. Iterative approximation of the model dynamics

The steady-state solution from section S3 above allows us to simulate the open ecosystem using the following iterative procedure:

1. Set $B = 0$, $P = 0$ and $t = 0$ ($t$ is the current time).
2. Draw a ($\gamma_i$, $\omega_i$) pair from a specified joint distribution $\Pr(\gamma, \omega)$ (which may be correlated or not).
3. If the pair fulfils $f(\gamma_i, \omega_i) > 0$, go to step 4 (computing densities), otherwise go to step 6 (advance time).
4. Re-compute $B$, all $b_i$s, and $P$ based on the set of ($\gamma$, $\omega$), using electronic supplementary material, equations (S21), (S14) and (S22).
5. If any ($\gamma_i$, $\omega_i$) pair no longer fulfils $f(\gamma_i, \omega_i) > 0$, then remove the pair with the smallest $b_i$. Go back to step 4 (recomputing densities).
6. Add $T$ to the current time $t$. Go to back to step 2 (add a new RM system).

This iterative procedure is an approximation of the full dynamics, but it is a good one whenever the time $T$ is large enough that the system reaches close to a steady state before the addition of each new strain. Electronic supplementary material, figures S2 and S3 show evidence that the diversity predicted by the iterative model matches the full dynamical model with $T = 10^4$ very well.

## 2.5. Implementation and data availability

We implemented our algorithm in Matlab (R2018b version 9.5.0.944444) and used the ODE solver ODE45 to solve the dynamical equations. The code and generated data files are available at the online repository located here [17]: https://github.com/RasmusSkytte/CorrelatedRestrictionModification Systems/tree/v1.0

# 3. Results

## 3.1. Correlated restriction–modification systems enable higher diversity

We first examine an open ecosystem model, where at periodic intervals of time a new bacterial strain with a unique RM strain is introduced, along with the corresponding phage variant (see Methods and materials section for details). The interval between each such injection is chosen to be long enough for the system to reach close to steady state. These conditions correspond to a simple view of the evolution of bacteria–phage ecosystems where, over longer timescales, novel RM systems evolve, or invade, and influence the existing ecosystems.

When the RM system strengths, $\omega_i$, and the bacterial growth rates, $\gamma_i$, are uncorrelated, we find, as did [13], that the diversity is limited by the burst size (figure 2*b*, light blue line). This limit was explained in [13]—the effective cooperation between bacteria results in the bacterial densities becoming roughly equal, and this, combined with the requirement that at least one phage on average survives from each burst to successfully infect a bacterium, means the number of strains cannot exceed the burst size of the phage. Indeed, we find that in this case, the bacterial density distribution exhibits a narrow peak (figure 2*c*, hatched bars).

However, when we introduce correlations by making the bacterial growth rates inversely related to the strength of their RM systems, then we find that the diversity can break through this burst size limit and increase up to twofold beyond it (figure 2*b*, dark blue line). Simultaneously, the equipartition of bacterial density is violated; the bacterial density distribution becomes much broader (figure 2*c*, shaded bars).

This broadening in the distribution of bacterial densities means that bacterial strains can coexist with competing strains despite their competitors being several orders of magnitude more abundant.

Intuitively, such a broad distribution of bacterial density (in particular the possibility for bacterial strains with quite low densities to coexist with other strains with much higher densities) is crucial for obtaining such high diversity.

The introduction of correlations also changes the final distribution of growth rates and RM strengths (figure 2d,e). In the uncorrelated case (hatched bars), the growth rates, $\gamma$, are narrowly distributed near the maximal value of 1, while the strength of the RM systems are broadly distributed between the smallest possible value ($10^{-4}$) and the highest value at which RM systems provide utility ($1/\beta$). When correlations are introduced (solid bars), we see a broader distribution of growth rates, albeit still biased towards faster growth. Interestingly, we observe a narrowing of the distribution of RM strengths whereby highly effective RM systems are no longer observed and bacteria with weaker RM systems can survive.

Note that the presence of the phage is crucial to obtain this high diversity. In the absence of the phage, competitive exclusion would result in only the bacterial strain with the highest growth rate surviving.

## 3.2. Explanation for increased diversity with correlated restriction–modification systems

The key to understanding why correlations give rise to more diversity is found by considering which combinations of costs and efficacy are possible for the RM systems. Mathematically, we can consider each RM system to lie somewhere in the $\gamma$–$\omega$ plane. The location of the RM systems in this plane will be constrained by physical limits—e.g. the growth rate $\gamma$ cannot be negative. We show the $\gamma$–$\omega$ plane schematically in figure 2f, where the added strains are picked from the enclosed region which depicts the set of possible values determined by the probability distribution $\Pr(\gamma, \omega)$.

From the steady state solution to the model (see §2.4 in Methods and materials and electronic supplementary material, equation (S22) in electronic supplementary material, section S3), we know that in order for a strain to be viable, it must fulfil the condition $f(\gamma_i, \omega_i) = \gamma_i(1 - B/C) - \alpha - \eta\omega_i P > 0$. This condition ensures that (i) the bacterial strain grows faster than their natural death rate ($\alpha$), and (ii) the combined growth rate and RM defence allows it to grow faster than it is being killed by phages.

Thus, for a strain to survive and coexist with the existing ecosystem, its $\gamma$ and $\omega$ values must lie in the green region, below and to the right of the red curve (which indicates $f(\gamma, \omega) = 0$). Conversely, a strain will go extinct if its $\gamma$ and $\omega$ values place it in the grey region above or to the left of the red curve since it will be killed faster than it can grow.

In turn, the successful addition of a strain causes the total bacterial density $B$ to increase, which results in the red line moving to the right. Thus, successive successful additions of new strains will eventually reduce the area of the viable (green) region that will result in survival. Further, electronic supplementary material, equation (S22) shows that if a strain survives, the closer it is to the red line, the lower its steady-state biomass.

In the absence of correlations, the distribution $\Pr(\gamma, \omega)$ uniformly covers a rectangle in the $\gamma$–$\omega$ space, as shown in figure 2g. In such a case, the red line typically settles to the far right, resulting in the surviving species having a narrow distribution of large growth rates, close to 1.

In contrast, when we introduce a trade-off, the shape of the grey region changes because of the correlation between the RM strengths and the growth rates, as shown in figure 2h. Here, the red line is unable to move as far to the right as in the previous case, since the points are closer to the red line and thus contribute less biomass to the system. The red line typically settles in a region that allows a broader distribution of growth rates in the surviving strains. Hence, one obtains a broader distribution of bacterial densities and a higher diversity.

By viewing the dynamics in $\gamma$–$\omega$ space, we can now understand why correlated RM systems change the distribution of growth rates and RM strengths (figure 2d,e). Without the correlations, strong RM systems are available for all growth rates and they drive the bacteria to a narrow distribution of growth rates. With correlations, the RM systems are less likely to simultaneously have both strong RM systems and a large growth rates. That is, there is a distinct lack of low $\omega$–high $\gamma$ species which in turn leads to more species coexisting and a broader distribution of growth rates.

We can understand this increased diversity intuitively if we modify the argument used in [13]: in the steady state, at least one phage from a burst must find a new host to propagate on or it would go extinct. This may be the host which the phage is epigenetically protected against, or it may be any of the ($D$–1) other bacterial strains in the system if the phage can escape restriction during the infection. The probability for a phage of variant $j$ to successfully propagate after adsorbing is then:

$$p_j = \frac{b_j}{B} + \sum_{i \neq j}^{D} \omega_i \frac{b_i}{B}. \tag{3.1}$$

For at least one phage per burst to successfully reproduce requires $p_j \geq 1/\beta$. This must be true for all phages in the system, so we can construct the aggregate statement by summing over all $D$ epigenetic variants:

$$\frac{D}{\beta} \leq \sum_{i=1}^{D} \frac{b_i}{B} + (D-1) \sum_{i=1}^{D} \omega_i \frac{b_i}{B} = 1 + (D-1) \sum_{i=1}^{D} \omega_i \frac{b_i}{B}. \tag{3.2}$$

If we rearrange this statement we get:

$$\frac{D}{\beta} \leq \frac{1 - \sum_{i=1}^{D} \omega_i (b_i/B)}{1 - \beta \sum_{i=1}^{D} \omega_i (b_i/B)}. \tag{3.3}$$

This expression highlights that if the RM systems all were perfect ($\omega_i \sim 0$), the diversity is indeed limited by $\beta$, and each bacterial strain reaches a population of $\delta/(\eta(\beta-1))$. Notably, If the efficacies of the RM systems are small enough, the phages begin to reliably propagate by attacking the hosts that are not epigenetically protected against them. This, in turn, leads to stronger suppression of the strains with the weaker RM systems (large $\omega$s) which broadens the biomass distribution and allows new bacterial strains to claim the unoccupied space in the ecosystem.

## 3.3. The effect of parameters on diversity

To explore how robust our results about diversity are to varying the different parameter values, we introduce a simplification of the dynamical model above to save computational time. Instead of simulating the entire set of dynamical equations, we evolve the ecosystem in the following iterative way:

1. Add bacterial strain with growth rate, $\gamma_i$, and RM strength, $\omega_i$, chosen from a given probability distribution $\Pr(\gamma, \omega)$.
2. Compute steady state.
3. Remove non-viable strains.
4. Go to step 1.

More details are provided in the Methods and materials section, but essentially this step-by-step process is an approximate way of determining which added strains will survive and which will go extinct at any given point in time. This procedure is an approximate representation of the full dynamics. However, it is a very good approximation when the interval between additions is large (see electronic supplementary material, figures S2 and S3).

### 3.3.1. The influence of $\Pr(\gamma, \omega)$

The 'steepness' or slope of the cloud of points in figure 2$h$ is an indication of the average cost to a bacterium of increasing the efficacy of its RM system. In reality, even very efficient RM systems reduce the growth rate of a bacterium by at most a few per cent [9], which would correspond to a much steeper cloud of points than we considered until now.

Therefore, in figure 3$a$, we decrease the average cost of having an RM system as described in §2.3 in Methods and materials. Note that despite the large change in the slope, the underlying correlations do not change substantially (see electronic supplementary material, figure S1A). The results are surprisingly robust to such changes and, despite the now smaller variation in growth rates, the system retains the increased diversity and the high bacterial biomass, while the phage population does drop substantially as the distribution steepens.

Next, we test how the correlations alone change the outcome of the simulation while we keep the average cost and efficacy constant. In figure 3$b$, we now draw the ($\gamma$, $\log 10(\omega)$) values from a multivariate normal distribution where we change the covariance between the random variables via the parameter $\rho$ such that when $\rho = 0$, the distributions are uncorrelated, and when $\rho = 1$ the distributions are fully correlated.

In this imagined case, the ecosystem changes drastically as the correlations are increased. When uncorrelated ($\rho \sim 0$), we recover the diversity limit of $D \sim \beta$ as from figure 2$b$, but as the correlations increase we observe a drastic increase in the diversity which reaches more than five times the burst size. Interestingly, this increase in diversity is accompanied by only a moderate increase in the phage population.

From these tests and the analytical solution we speculate that it is the limited access to high $\gamma$–low $\omega$ pairs that are key to high diversity ecosystems. In fact, we find that even when in the uncorrelated RM system case, we can still obtain diversity higher than the burst size when we no longer allow for strong

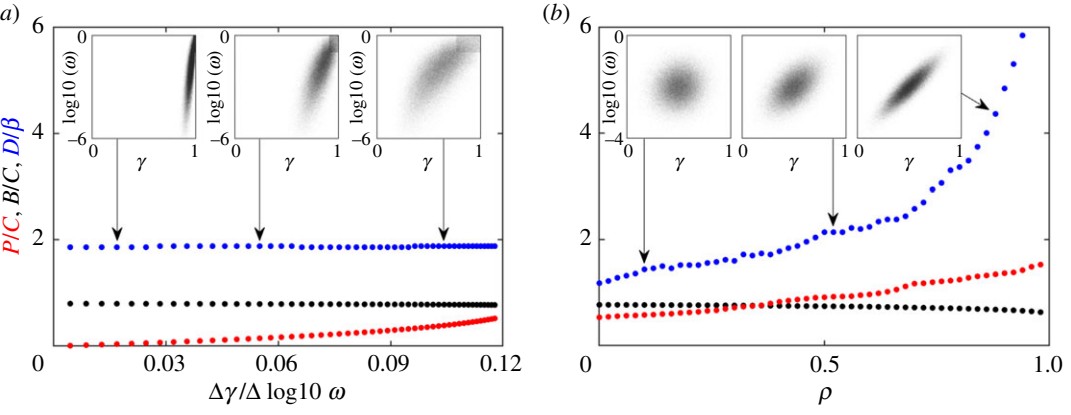

**Figure 3.** Distribution shape and diversity. We vary the $\gamma$–$\omega$ distributions and use the iterative algorithm to predict the final populations. (*a*) By limiting the cost of individual RM systems, we change the overall slope of the Pr($\gamma$, $\omega$) distribution without changing the correlations. (*b*) Using a joint normal distribution for $\gamma$ and log10($\omega$) can change from an uncorrelated distribution to a fully correlated one by increasing $\rho$ without changing the mean values. (*a,b*) Insets indicate the sampled values of ($\gamma$, $\omega$) at the indicated locations.

RM systems—in such a case the bacterial density distribution becomes broad (see electronic supplementary material, figure S4).

## 3.4. The effect of phage and bacterial parameters on diversity

In figure 4, we use the iterative algorithm to investigate how the diversity varies as we vary the bacterial death rate $\alpha$, the phage burst size $\beta$, the phage decay rate $\delta$, and the phage adsorption rate $\eta$.

Notice that the values we plot are final values obtained by the simulated evolution after adding $10^5$ species. These values are not the steady state values, as adding more species would change the results. What the graph does show, is the relative difficulty for the ecosystems of bacteria to overcome the phage predation. We see that the diversity and population levels will vary as we change the parameters, but in general it remains very stable across a wide range of parameters. In this stage, the phages are suppressed to low levels and the bacteria are close to their maximal population size while retaining a high diversity of ~2$\beta$. From figure 4*c*, we see that as the bacterial death rate increases, a corresponding loss of bacterial biomass and diversity follows, since the bacteria need higher growth rates to compensate for the increased death rate.

The last three panels (figure 4*a,b,d*) show varying phage parameters. In general, we can consider these changes as either an increase in virulence or decreases in virulence. For example, the virulence decreases as the phage death rate increases (figure 4*a*), the adsorption rate decreases (figure 4*b*), or when the burst size decreases (figure 4*d*). For the death rate and the adsorption rate, the results are similar: as the phage becomes less potent, the bacteria are suppressed less and begin to compete with each other causing a decrease in the diversity through increased competitive exclusion. This is not observed when the burst size decreases, but note that diversity is measured in terms of the burst size and therefore diversity *does* drop but not more than expected ($D \sim 2\beta$).

In contrast, we can consider what happens when the virulence increases by (figure 4*a*) decreasing phage death rates, (figure 4*b*) increasing adsorption rate, or (figure 4*d*) increasing burst size. Again the death rate change and the adsorption rate change yield similar results where the phage are less likely to decay before finding a bacterium to infect and thus they more strongly control the ecosystem and cement the high diversity state. The burst size of the phage does influence the stability of the ecosystem as the increase in burst size gives a smaller increase in diversity than expected. This is because the increases in phage pressure require stronger RM systems to overcome. Since the underlying probability distribution for Pr($\gamma$, $\omega$) does not change, it takes longer for fit bacteria to arise—making it harder for the bacteria to reach the high diversity state.

## 3.5. Strength of the restriction–modification system as a co-determinant of survival

In our simulations, e.g. figure 2*b*, bacteria can coexist despite different bacterial strains having up to 10% variation in growth rates (figure 2*d*). The question is now: why does this wide distribution not lead to the slower-growing bacteria being out-competed by competitive exclusion?

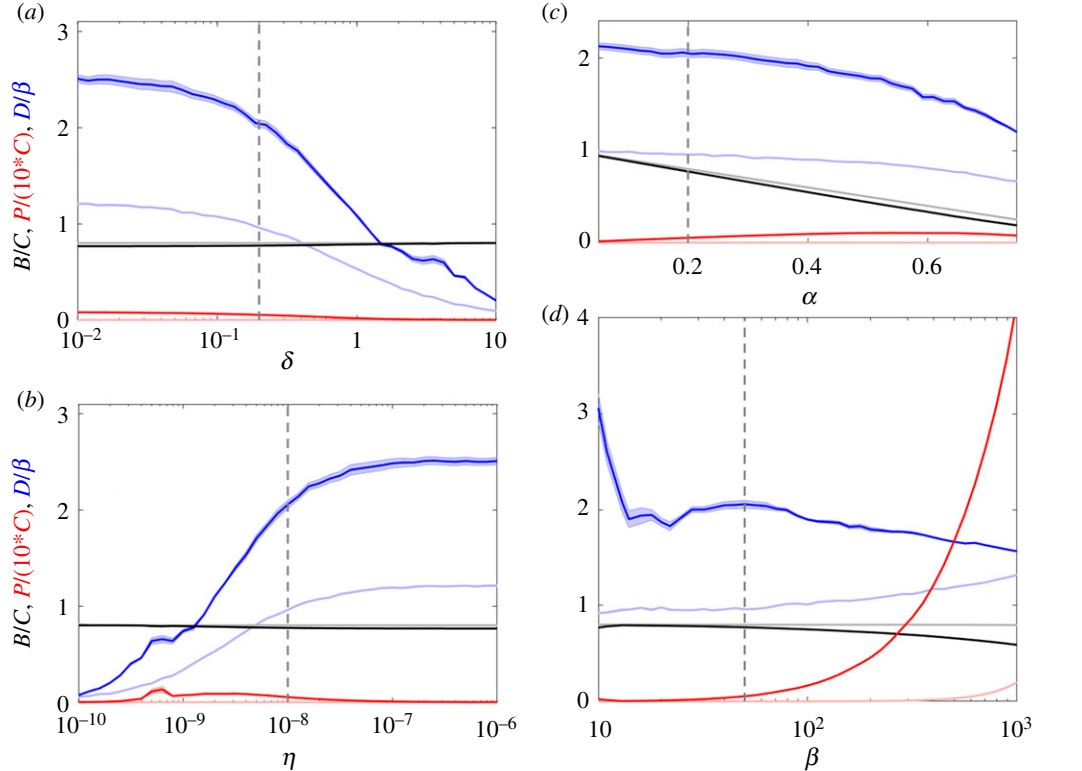

**Figure 4.** Diversity dependence on parameter values. We show the final populations and diversity after $10^5$ rounds of the iterative algorithm under varying parameters: (a) the phage decay rate, $\delta$, (b) the phage adsorption rate, $\eta$, (c) the bacterial death rate, $\alpha$ and (d) the phage burst size, $\beta$. Dark (light) colours indicate simulations using (un)correlated RM strengths and bacterial growth rates. Shaded areas indicate standard error of the mean over five replicates but this error is often too small to see on the figure. Dashed lines indicate the default parameter values.

One key observation from electronic supplementary material, equation (S22), is that the amount of biomass each point contributes to the whole increases not just with the bacterial growth rate, but also with increasing RM strength.

In figure 4, we plot the biomass of each bacterial strain as a function of (i) its effective growth rate $\gamma_i'$ (i.e. the rate of growth minus the rate of death), (ii) the strength of the RM systems $1/\omega_i$, and (iii) the ratio $\gamma_i'/\omega_i$. In figure 5a–c, we use data at the end of the simulated evolution (figure 2b) and in figure 5d–f, we use a simulated batch culture with 100 bacterial strains.

In both cases, we see that neither the effective growth rate nor the RM strength can fully explain the achieved biomasses of the bacterial strains, but that the combination of the two can. The batch culture is particularly interesting since it clearly shows that the strains with the highest growth rates and those with the strongest RM systems go extinct.

As one would expect, we see in the simulations that the effective growth rate must be positive for the bacteria to survive, but beyond that, the growth rate has little influence on the achieved biomass. For the surviving bacteria, the strength of the RM systems becomes a good predictor for the biomass but the product of the effective growth rate and the RM strength is an almost perfect predictor of the bacterial biomass.

# 4. Discussion

We have demonstrated that when there is a trade-off between bacterial growth rate and the efficacy of its RM defence system, a remarkably large number bacterial strains can coexist on the same resource. The work of [9] does provide evidence for such a trade-off and our simulations (figure 3a) indicate that even a weak trade-off of this kind is sufficient to produce coexistence of many strains.

We show that in these phage limited ecosystems, it is not the growth rate of the bacteria alone that determines whether they go extinct or not—this is determined instead by a combination of effective growth rate *and* the strength of the RM system (figure 5). Further, for strains that do not go extinct, this combination—$\gamma'/\omega$—determines their relative biomass.

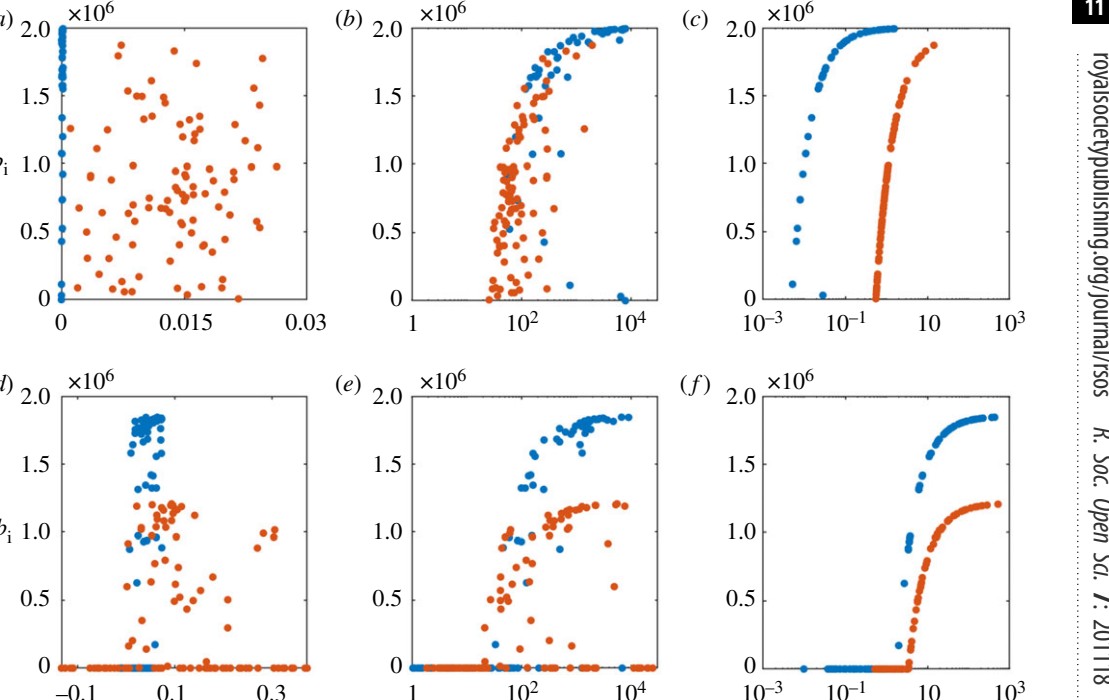

**Figure 5.** Predictors of survival. Using the data from the end of the simulated evolution, we show the population size as a function of (a) effective growth rates $\gamma_i' = \gamma_i(1 - B/C) - \alpha$, (b) strength of RM system $1/\omega_i$, and (c) ratio of effective growth rate $\gamma_i'$ to RM imperfection $\omega_i$. (d–f ) Same as (a–c) but using data from a simulated batch culture consisting of 100 bacterial strains. In all cases red (blue) points correspond to (un)correlated RM systems.

To be more precise, there exists a demarcation line in the growth rate versus RM efficacy plane (the red curve in figure 2). This 'extinction curve' separates strains who either grow too slow or have too weak RM systems from the strains than are able to survive in the ecosystem. Importantly, the shape of this curve depends on $\gamma'/\omega$ and not $\gamma$ alone, which means that in the region where the strains do not go extinct, the distance from the extinction curve (proportional to $\gamma'/\omega$) determines their biomass (as shown in figure 5). In our model ecosystem, the distance may thus be thought of as a measure of 'fitness' of a bacterial strain, and $\gamma'/\omega$ acts as a quantitative (albeit nonlinear) measure of fitness.

It would be interesting to design experimental tests for our predictions for phage–bacteria ecosystems where the bacteria have RM defences. Specifically, we predict: (i) the number of strains that can coexist will be well over the burst size of the phage, (ii) the fitness of bacterial strains is a function of the ratio of the growth rate to the RM imperfection, rather than the growth rate alone, and (iii) there exists a specific 'extinction line' that separates the growth rate–RM imperfection space into regions where strains can and cannot survive. One can imagine co-culturing bacterial strains that have been engineered from the same parent strain with different known RM systems, to test these predictions. It is important to note that the selection pressure on the cost and efficacy of the RM systems is dependent on the richness of the ecosystem. That is, the selection pressure increases as the total bacterial density $B$ nears its maximal value. Therefore, many bacterial strains (of the order of the burst size of the phage) would need to be co-cultured to test these predictions. This, however, presents an experimental challenge since co-culturing bacterial strains, even in moderate numbers, is often difficult and could introduce additional interactions that our model does not account for. In practice, this makes it experimentally difficult to validate the first prediction with typical phage that have burst sizes of the order of several tens if not hundreds. Therefore, it might be more suitable to work with mutant phage that, if possible, have particularly low burst sizes. The other two predictions are expected to be more compatible with experimental designs, since selection here can be made stronger by artificially increasing the cost of the RM systems, for example by introducing a costly mutation alongside the RM system, rather than attempting to reach high ecosystem diversity.

There are several other systems where such trade-offs between growth and defence control the fitness: One more example from the world of bacteria is when bacteria are repeatedly exposed to periods of high nutrient availability followed by periods of starvation [18]; another example, in plants, is described in [19].

Our results mirror the lessons learned from studies of coexistence of species, in particular how the partitioning of predators can have a stabilizing effect on a competitive ecosystem [20]. Since the bacterial strains in our analysis are highly similar, resource partitioning is assumed to be negligible, and thus it is the strong partitioning of predators generated by the RM systems that causes substantial niche differentiation and allows for high diversity. We speculate that this effect is enhanced in our system because the predator changes type when preying on its secondary targets. This results in a negative feedback loop such that when a prey (bacterial host) grows in abundance, it is preyed on more by both its secondary predators and primary predators (phage of different epigenetic types) as expected. However, the predation turns the secondary predators into primary predators which then strongly inhibit the prey. Thus, in our system, *intra*species interactions result in a strong *inter*species interaction—an interesting twist on the predator partitioning discussed in [20].

We further show that such high diversity ecosystems are quite robust to the parameters of the phage, but that this state starts breaking down when the phage is not virulent enough to control the bacterial biomass (figure 4). In such cases, the ecosystems are not phage-limited but rather resource-limited and the competitive exclusion drives the system towards a winner-takes-all ecosystem. This behaviour is similar to what is found in the discrete spatial model of [12], where the phages almost die out and cannot control the bacterial biomass. Here they find, as we do, that the diversity of the ecosystem is very limited.

The increased difficulty with reaching high diversity ecosystems observed at some parameter values is consistent with the 'narrowing staircase of coexistence' discussed in [21]. In our system, the 'narrowing' of the sustainable parameter space is explicitly visualized by the red line in the $\gamma$–$\omega$ space as it moves down and to the right over time. The gradual evolution of the ecosystems drives increasing constraints on the parameters of new bacterial strains.

In their paper [22], the authors suggest that the wide distribution of growth rates found in nature can be explained by a receptor-based trade-off between nutrient uptake (favouring receptors) and phage infections, as phage often use nutrient receptors as sites of recognition and infection. The bacterium can thus limit its phage exposure at the cost of a loss of growth rate by expressing fewer receptors. Here we demonstrate an alternative mechanism by which correlations in $\gamma$–$\omega$ space can widen the distribution of growth rates. Since RM systems are very efficient, even very small correlation yields large reduction in phage exposure.

Bacteria typically evolve resistance to phages by loss of a receptor, and while such resistance gives perfect protection against a phage variant, the loss of receptor causes a substantial loss in growth rate [23]. In our framework, this can be understood as a zero-$\omega$–small-$\gamma$ strain, and in accordance with what [11] found, such strains will be out-competed in a sufficiently diverse ecosystem. This highlights that, in comparison with reducing receptor expression levels, RM systems may be able to achieve similar efficacy of defence against a phage but at a much smaller cost.

Furthermore, when communities of several species coexist, several synergistic mechanisms such as cross-feeding may come into play that may counteract the growth-rate loss of investing in the RM system [24–26].

RM systems are believed to influence the speciation of bacteria due to their ability to control horizontal gene transfer and thus genetically isolate the bacteria [7]. In addition, it is believed that RM systems diversify the genetic pool by promoting DNA recombination [9]. From the perspective of evolution, in the ecosystems that develop in our model, we may speculate that the high diversity allows for more genetic variation, and thus presumably a faster rate of evolution, while the wide distribution in growth rates gives more room for bacteria to evolve, e.g. by reciprocal sign epistasis [27].

We end by noting again that the life of bacteria is a harsh life, and that they exist in a constant stressful environment, be that stress stemming from predation, from starvation, or from toxins/antibiotics, or abiotic factors. Our work reinforces the idea that in order to understand what constitute 'fit' bacteria in a realistic environment, one must consider the trade-off between growth and defence, or growth and strategies to alleviate other stressors.

Data accessibility. Our data is available in the Zenodo repository: https://doi.org/10.5281/zenodo.3842682.
Authors' contributions. R.S.E. and S.K. designed the study, developed the theoretical formalism and carried out the analytical calculations. R.S.E. implemented and analysed the simulations. R.S.E. and S.K. wrote the manuscript.
Competing interests. We declare we have no competing interests.

Funding. R.S.E. has received funding for this project from the European Research Council (ERC) under the European Union's Horizon 2020 research and innovation programme under grant agreement no. 740704. S.K. thanks the Simons Foundation for funding, as well as the Department of Atomic Energy, Government of India, under project no. 12-R&D-TFR-5.04-0800.

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
