## [Reviewer comments · Royal Society Open Science]

Review History

RSOS-201118.R0 (Original submission)

Review form: Reviewer 1

Is the manuscript scientifically sound in its present form?

Yes

Are the interpretations and conclusions justified by the results?

Yes

Is the language acceptable?

Yes

Do you have any ethical concerns with this paper?

No

Have you any concerns about statistical analyses in this paper?

No

Recommendation?

Accept with minor revision (please list in comments)

Comments to the Author(s)

Defense versus growth in a hostile world - Lessons from phage and bacteria

By Rasmus Skytte Eriksen & Sandeep Krishna

This manuscript extends the results of the model by K. Sneppen et al (ref 13), which is co-authored by one of the authors of this paper. Here, the authors explore the effects of correlations between bacterial growth rate and the rate of failures of the RM immune system in phage backgrounds. They find that these cost-benefit tradeoffs lead to a greater diversity threshold than that found in ref. 13. Their results intuitively make sense and are backed by modeling and simulations. Overall, I find the results sufficiently sound and interesting to recommend publication provided following comments are addressed in a revision:

In the introduction, more concrete details on costs in RM systems and how these couple to “strength” are needed. References to known results (experimental and theoretical) would greatly strengthen the argument presented in L14-17 on p2. For example, how does growth rate depend on RM failure rate; would there be a higher fitness cost of maintaining a robust RM system and where would these costs come from? Would RM systems which more frequently fail also have larger self-targeting costs, etc? Also, explicitly state to which extent are the correlations between gamma and omega supported by known results and to which extent are they exploratory.

From the modeling perspective, the main difference in the model wrt ref. 13 is the inclusion of carrying capacity C. To what extent does C impact the results of the paper, since the equations allow measuring the concentration of bacteria in units of C? The value for C used maintains large populations, where fitness effects are dominant. When C is reduced, does this limit the extent of sustained diversity? Is there a transition between $D \sim \beta$ and higher diversity regime as C is varied? A more detailed discussion of these effects for large/small populations could help place this work in an appropriate regime.

Minor comments:

P1 L56 consider using efficacy instead of efficiency, and in the text thereafter. “Efficiency” already includes the costs, therefore the term “cost-efficiency” seems unnecessary.

P3 L38-49 consider defining P and B; I am assuming that $B = \sum_i b_i$, and similarly for P, but it would be helpful to have that written explicitly in the main text.

Review form: Reviewer 2

Is the manuscript scientifically sound in its present form?

Yes

Are the interpretations and conclusions justified by the results?

Yes

Is the language acceptable?

Yes

Do you have any ethical concerns with this paper?

No

Have you any concerns about statistical analyses in this paper?

No

Recommendation?

Accept with minor revision (please list in comments)

Comments to the Author(s)

From my perspective, this paper is a worthwhile contribution to the journal. I would like to see two main changes but I think they are minor in terms of work and more in terms of general formatting. I would like the introduction to spend a little more time laying out the burst size and diversity link, the authors have prior art on this and maybe a crash course in the introduction of the logic would better frame the paper. When the authors cite ref 13 I think this would be a good place to more broadly talk about the logic here. My other comment would be to try and simplify the language for non-mathematics people, myself included, as a phage biologist I read and followed, mostly, with some great difficulty. I acknowledge that the authors are dealing with non-rudimentary ideas (extinction curves in the growth and RM planes) but for this paper to have real value to experimentalists like myself I need to be able to synthesize how I can better test the predictions of the models. Questions like, if I test bacteria with different growth rates and RM strength what would I predict given this model.

Decision letter (RSOS-201118.R0)

Dear Dr Krishna

On behalf of the Editors, I am pleased to inform you that your Manuscript RSOS-201118 entitled "Defense versus growth in a hostile world - Lessons from phage and bacteria" has been accepted for publication in Royal Society Open Science subject to minor revision in accordance with the referee suggestions. Please find the referees' comments at the end of this email.

The reviewers and handling editors have recommended publication, but also suggest some minor revisions to your manuscript. Therefore, I invite you to respond to the comments and revise your manuscript.

- Ethics statement

- Data accessibility

<http://datadryad.org/submit?journalID=RSOS&manu=RSOS-201118>

- Competing interests

- Authors' contributions

- Acknowledgements

- Funding statement

Because the schedule for publication is very tight, it is a condition of publication that you submit the revised version of your manuscript before 07-Aug-2020. Please note that the revision deadline will expire at 00.00am on this date. If you do not think you will be able to meet this date please let me know immediately.

If your manuscript is newly submitted and subsequently accepted for publication, you will be asked to pay the article processing charge, unless you request a waiver and this is approved by Royal Society Publishing. You can find out more about the charges at <https://royalsocietypublishing.org/rsos/charges>. Should you have any queries, please contact openscience@royalsociety.org.

on behalf of Professor Matjaz Perc (Associate Editor) and Pete Smith (Subject Editor)
openscience@royalsociety.org

Reviewer comments to Author:

Reviewer: 1
Comments to the Author(s)

Defense versus growth in a hostile world - Lessons from phage and bacteria
By Rasmus Skytte Eriksen & Sandeep Krishna

This manuscript extends the results of the model by K. Sneppen et al (ref 13), which is co-authored by one of the authors of this paper. Here, the authors explore the effects of correlations between bacterial growth rate and the rate of failures of the RM immune system in phage backgrounds. They find that these cost-benefit tradeoffs lead to a greater diversity threshold than that found in ref. 13. Their results intuitively make sense and are backed by modeling and simulations. Overall, I find the results sufficiently sound and interesting to recommend publication provided following comments are addressed in a revision:

In the introduction, more concrete details on costs in RM systems and how these couple to “strength” are needed. References to known results (experimental and theoretical) would greatly strengthen the argument presented in L14-17 on p2. For example, how does growth rate depend on RM failure rate; would there be a higher fitness cost of maintaining a robust RM system and where would these costs come from? Would RM systems which more frequently fail also have larger self-targeting costs, etc? Also, explicitly state to which extent are the correlations between gamma and omega supported by known results and to which extent are they exploratory.

From the modeling perspective, the main difference in the model wrt ref. 13 is the inclusion of carrying capacity C. To what extent does C impact the results of the paper, since the equations allow measuring the concentration of bacteria in units of C? The value for C used maintains large populations, where fitness effects are dominant. When C is reduced, does this limit the extent of sustained diversity? Is there a transition between $D \sim \beta$ and higher diversity regime as C is varied? A more detailed discussion of these effects for large/small populations could help place this work in an appropriate regime.

Minor comments:

P1 L56 consider using efficacy instead of efficiency, and in the text thereafter. “Efficiency” already includes the costs, therefore the term “cost-efficiency” seems unnecessary.

P3 L38-49 consider defining P and B; I am assuming that $B = \sum_i b_i$, and similarly for P, but it would be helpful to have that written explicitly in the main text.

Reviewer: 2

Comments to the Author(s)

From my perspective, this paper is a worthwhile contribution to the journal. I would like to see two main changes but I think they are minor in terms of work and more in terms of general formatting. I would like the introduction to spend a little more time laying out the burst size and diversity link, the authors have prior art on this and maybe a crash course in the introduction of the logic would better frame the paper. When the authors cite ref 13 I think this would be a good place to more broadly talk about the logic here. My other comment would be to try and simplify the language for non-mathematics people, myself included, as a phage biologist I read and followed, mostly, with some great difficulty. I acknowledge that the authors are dealing with non-rudimentary ideas (extinction curves in the growth and RM planes) but for this paper to have real value to experimentalists like myself I need to be able to synthesize how I can better test the predictions of the models. Questions like, if I test bacteria with different growth rates and RM strength what would I predict given this model.

Author's Response to Decision Letter for (RSOS-201118.R0)

See Appendix A.

Decision letter (RSOS-201118.R1)

Dear Dr Krishna,

It is a pleasure to accept your manuscript entitled "Defence versus growth in a hostile world - Lessons from phage and bacteria" in its current form for publication in Royal Society Open Science.

on behalf of Professor Matjaz Perc (Associate Editor) and Pete Smith (Subject Editor)
openscience@royalsociety.org

Appendix A

Dear Editors and reviewers,

We would like to thank you and the reviewers for your comments in regards to our manuscript.

Based on your feedback we have made a number of changes to our manuscript and we believe the manuscript is the better for it. Below, we respond to the comments from the reviewers on a point-by-point basis.

We believe this revised version addresses the concerns raised by the reviewers.

Sincerely,
Rasmus Skytte Eriksen and Sandeep Krishna.

Response to reviewer 1

In the introduction, more concrete details on costs in RM systems and how these couple to “strength” are needed. References to known results (experimental and theoretical) would greatly strengthen the argument presented in L14-17 on p2. For example, how does growth rate depend on RM failure rate; would there be a higher fitness cost of maintaining a robust RM system and where would these costs come from? Would RM systems which more frequently fail also have larger self-targeting costs, etc? Also, explicitly state to which extent are the correlations between gamma and omega supported by known results and to which extent are they exploratory.

We thank the reviewer for pointing this out and agree that clearer statements of known results, and the nature of our explorations, was needed. We have added a paragraph to the Introduction that states:

Presently, the relationship between the cost and efficacy of RM systems has not been determined with quantitative precision. However, it is known that such costs exist, although they are small - RM systems with high efficacy seemingly impose only a few per cent penalty on the growth rate for bacteria. Such costs can be observed in competition assays between strains with and without RM systems, but are hard to detect in bacterial doubling rates. Mechanistically, RM defences have been speculated to give a fitness cost, e.g., some RM systems require large amounts of ATP during restriction. It has also been shown that some RM systems may act as autoimmune systems that cause DNA damage on the strains containing them. These costs seemingly imply that RM systems impose a cost on their host which is, at least partially, correlated to their efficacy.

In the revised manuscript, this paragraph also contains references to work that supports the statements we have made. We hope this addresses the reviewer’s concerns.

From the modeling perspective, the main difference in the model wrt ref. 13 is the inclusion of carrying capacity C . To what extent does C impact the results of the paper, since the equations allow measuring the concentration of bacteria in units of C ? The value for C used maintains large populations, where fitness effects are dominant. When C is reduced, does this limit the extent of sustained diversity? Is there a transition between D beta and higher diversity regime as C is varied? A more detailed discussion of these effects for large/small populations could help place this work in an appropriate regime.

The reviewer is indeed correct that the behaviour of the equations is unaffected by using a value of C different from 1, because one can always measure populations in units of C (thereby rescaling all bacterial and phage populations, as well as the parameter η). It also makes no difference when we simulate an open ecosystem, where we choose a threshold value at which a bacterial strain (or phage) is said to have gone extinct and is removed from the ecosystem. Again, choosing carrying capacity to be C and the threshold to be τ is identical to choosing the carrying capacity to be 1 and the threshold to be τ/C . However, the reviewer is indeed correct that our results might change if we vary the carrying capacity while keeping the threshold constant (which would be like saying a strain goes extinct when it drops below a single individual, while varying the maximum number of bacteria that can be present in the ecosystem.) For a small enough population size, we expect there to be significant stochasticity due to the small numbers and it would be more proper to perform simulations using stochastic versions of the equations. In our study, since we perform only deterministic simulations, we are assuming that we are firmly within the large population regime. Our choice of $C = 10^8$ and extinction threshold $\tau = 1$ are consistent with this. We hope to explore stochastic effects in a future work, but we included additional comments in the revised manuscript on page 3 in lines 20-23 to explain these issues and clarify that we are examining the large population regime.

P1 L56 consider using efficacy instead of efficiency, and in the text thereafter. "Efficiency" already includes the costs, therefore the term "cost-efficiency" seems unnecessary.

We have changed the language accordingly throughout the manuscript since we agree that the word "efficacy" more accurately conveys our meaning. We thank the reviewer for this suggestion.

P3 L38-49 consider defining P and B ; I am assuming that $B = \sum_i b_i$, and similarly for P , but it would be helpful to have that written explicitly in

the main text.

We have now done so. This should help the readability of the manuscript. Furthermore, we have changed the name of our joint probability distribution $P(\gamma, \omega)$ to $\Pr(\gamma, \omega)$ to avoid confusion between it and the total phage population.

Response to reviewer 2

I would like the introduction to spend a little more time laying out the burst size and diversity link, the authors have prior art on this and maybe a crash course in the introduction of the logic would better frame the paper. When the authors cite ref 13 I think this would be a good place to more broadly talk about the logic here.

We thank the reviewer for their very useful comments. We have expanded the explanation of the burst size - diversity link in the Introduction on page 2 between lines 13 and 24. We hope this makes the logic clearer.

My other comment would be to try and simplify the language for non-mathematics people, myself included, as a phage biologist I read and followed, mostly, with some great difficulty. I acknowledge that the authors are dealing with non-rudimentary ideas (extinction curves in the growth and RM planes) but for this paper to have real value to experimentalists like myself I need to be able to synthesize how I can better test the predictions of the models. Questions like, if I test bacteria with different growth rates and RM strength what would I predict given this model.

We completely agree that the paper is difficult to read because we go back and forth between mathematical and verbal arguments. We have tried to simplify this by confining the more mathematical arguments to certain clearly demarcated sections. We perhaps have not completely succeeded in making it easier, so we have also added a paragraph to the Discussion section which attempts to more simply summarize our results and specifically put them in the context of possible experiments. We hope that in combination these changes will make it easier to read for non-mathematical readers. We thank the reviewer for highlighting these instances where our writing is less accessible than we would like and hope our changes makes the paper more readable for the broader readership.